# Cochrane systematic review and meta-analysis of botulinum toxin for the prevention of migraine

Clare P Herd, [1] Claire L Tomlinson,[2] Caroline Rick,[3] William J Scotton,[4] Julie Edwards,[5] Natalie J Ives,[2] Carl E Clarke,[6] AJ Sinclair[4]

► Additional material is published online only. To view please visit the journal online (http://dx.doi.org/10.1136/bmjopen-2018-027953)

[1] Institute of Applied Health Research, University of Birmingham, Birmingham, UK
[2] BCTU, University of Birmingham, Birmingham, UK
[3] Nottingham Clinical Trials Unit, University of Nottingham, Nottingham, Nottinghamshire, UK
[4] University of Birmingham, Institute of Metabolism and Systems Research, Birmingham, UK
[5] Sandwell and West Birmingham Hospitals NHS Trust, Department of Neurology, Birmingham, Birmingham, UK
[6] University of Birmingham, Neurology, Birmingham, UK

**Correspondence to**
Dr AJ Sinclair;
a.b.sinclair@bham.ac.uk

## ABSTRACT

**Objectives** To assess the effects of botulinum toxin for prevention of migraine in adults.

**Design** Systematic review and meta-analysis.

**Data sources** CENTRAL, MEDLINE, Embase and trial registries.

**Eligibility criteria** We included randomised controlled trials (RCTs) of botulinum toxin compared with placebo, active treatment or clinically relevant different dose for adults with chronic or episodic migraine, with or without the additional diagnosis of medication overuse headache.

**Data extraction and synthesis** Cochrane methods were used to review double-blind RCTs. Twelve week post-treatment time-point data was analysed.

**Results** Twenty-eight trials (n=4190) were included. Trial quality was mixed. Botulinum toxin treatment resulted in reduced frequency of −2.0 migraine days/month (95% CI −2.8 to −1.1, n=1384) in chronic migraineurs compared with placebo. An improvement was seen in migraine severity, measured on a numerical rating scale 0 to 10 with 10 being maximal pain, of −2.70 cm (95% CI −3.31 to −2.09, n=75) and −4.9 cm (95% CI −6.56 to −3.24, n=32) for chronic and episodic migraine respectively. Botulinum toxin had a relative risk of treatment related adverse events twice that of placebo, but a reduced risk compared with active comparators (relative risk 0.76, 95% CI 0.59 to 0.98) and a low withdrawal rate (3%). Although individual trials reported non-inferiority to oral treatments, insufficient data were available for meta-analysis of effectiveness outcomes.

**Conclusions** In chronic migraine, botulinum toxin reduces migraine frequency by 2 days/month and has a favourable safety profile. Inclusion of medication overuse headache does not preclude its effectiveness. Evidence to support or refute efficacy in episodic migraine was not identified.

## Strengths and limitations of this study

► This paper is a summary of a Cochrane review conducted using systematic and thorough methodology to identify and synthesise all available evidence for the effectiveness of botulinum toxin for prophylactic treatment of migraine.

► No language or date restrictions were placed on the search strategy.

► Many of the included studies were small in size and failed to fully report their data which impacted the quality ratings and the content of the meta-analyses.

► Our chosen primary outcome measure, though recommended in current guidelines for controlled trials of prophylactic treatment of chronic migraine, was not commonly recorded.

Botulinum toxin type A (BTX-A) has been licensed for use in migraine in some countries, based largely on two commercially sponsored trials.[6 7] The recommended reconstituted dose is 155 to 195 units, administered intramuscularly as 0.1 mL (five units) injections to between 31 and 39 sites around the head and neck.[3] Cost of treatment and administration of BTX-A is much higher than standard doses of the two first line treatments for the prevention of migraine, propranolol and topiramate (around 25 times and 15 times respectively in the UK).[8–10]

Migraine can be categorised as chronic or episodic and these terms are commonly used in eligibility criteria for clinical trials and systematic reviews. Chronic migraine is currently defined by the International Headache Society (IHS) as headache for at least 15 days per month with migraine features on eight of those days.[11] Episodic migraine is commonly used to describe patients with symptoms of migraine who have less than 15 headache days per month and according to official guidance is a term which can be used for migraine that is not covered by the definition of chronic migraine.[11] Migraine can occur with medication overuse headache; the

## INTRODUCTION

Migraine is the seventh leading cause of years lived with disability globally and is estimated to affect around 15% of the world's population.[1] Days lost from work and other activities of daily living resulting from migraines have a major economical impact.[2] Many people with migraine suffer prolonged and frequent migraine attacks despite optimised acute and prophylactic treatments.[3–5]

IHS definition has evolved, but currently this is defined as an interaction between a therapeutic agent used excessively and a susceptible patient.[11 12] Trials recruiting participants with chronic migraine will come across many patients with this dual diagnosis. Current UK guidelines published by The National Institute for Health and Care Excellence (NICE) recommend the use of BTX-A for chronic migraine, but not for high frequency episodic migraine, and only when the condition is 'appropriately managed' for medication overuse.[8]

The aim of this evidence review was to assess the effects of botulinum toxin (BTX) versus placebo or alternative active treatment for the prophylaxis of episodic migraine or chronic migraine in adults.

This paper is a summary of key aspects from a Cochrane review first published in The Cochrane Library 2018, Issue 6 (see http://www.thecochranelibrary.com/ for information).[13] Cochrane reviews are regularly updated as new evidence emerges and in response to feedback, and The Cochrane Library should be consulted for the most recent version of the review.

## METHODS

The protocol for this review was published in the Cochrane Database of Systematic Reviews in advance of the publication of the full review which replaced it. Deviations from the protocol are listed in the full review.[13]

### Search strategy

A systematic search of the literature published before March 2019 was carried out. We designed a highly sensitive search strategy using methods recommended by the Cochrane collaboration to minimise publication bias. No date, language or publication status restrictions were applied. We used a combination of index terms and free text terms for headache, migraine, cephalalgia or hemicrania; and botulinum toxin, Botox, onabotulinum toxin, Oculinum or clostridium botulinum. Relevant trials were identified through electronic searches of Cochrane Central Register of Controlled Trials, MEDLINE (see full strategy in online supplementary file 1), Embase, ClinicalTrials.gov and WHO International clinical trials registry, hand-searching reference lists and citation searches on key publications and correspondence with all major manufacturers of BTX products relevant to this review.

### Selection criteria

We included randomised, double-blind, controlled trials of people over the age of 18 years suffering from migraine as defined by any edition of the IHS criteria,[11 12 14] or meeting reasonable criteria designed to distinguish between migraine and tension-type headache. Patients with both chronic migraine and episodic migraine were included in this review. Medication overuse headache was included as these types of participants have been included in large and prominent trials in this area. Trials must compare BTX (any sero-type) injected into the head and

neck muscles with placebo injections, clinically relevant different dose of same treatment or active preventative agent. Trials allowing the use of concomitant preventative or rescue treatments were included.

Screening of abstracts and assessment of eligibility of full papers were carried out independently in duplicate and according to criteria predefined in the peer reviewed protocol. If disagreements occurred at any stage, a third author considered the available information or if necessary the study authors were contacted for clarification. When eligibility could not be determined through consideration of published materials or contact with trial authors the studies were excluded.

### Quality assessment

Eligible material was assessed, independently by two reviewers for each trial, for methodological quality using Cochrane risk of bias methods. Publications were assessed on their method of randomisation, blinding and concealment of allocation, the number of participants lost to follow-up, evidence of selective reporting and study size.

We considered the use of funnel plots to assess the risk of publication bias but did not carry them out. We made this decision because of the small number of studies included in the individual meta-analyses and the true heterogeneity in the trial design (dose, injection paradigm) and populations studied (migraine sub-classifications), which would make it impossible to draw useful conclusions from the plots. Grading of Recommendations Assessment, Development and Evaluation (GRADE) tables were created for each comparison, this process involves assessment of the risk of publication bias for each outcome measure.

### Data extraction

Data extraction was carried out independently and in duplicate onto forms designed and tested at protocol stage. The primary outcome was frequency of migraine days per month. Secondary outcomes included: frequency of headache days, frequency of migraine attacks, severity of migraine, duration of migraine, 50% responder rate, global impression scales, quality of life measures and adverse event reporting. We used risk ratios (RRs) as the preferred statistical output for dichotomous outcomes, with 95% CIs. For continuous data, we used mean differences with 95% CIs. Results with p values lower than 0.05 were considered to be statistically significant. Twelve week time-point data following final round of treatment was analysed. We sought data from the first phase for any cross-over trials identified. We attempted to contact authors and obtain missing data.

### Statistical analysis

The review authors assessed trial information and baseline characteristics to identify clinical and methodological differences during the data extraction process. If clinical and methodological homogeneity were confirmed, we carried out meta-analysis of the data using Review Manager (RevMan) V.5.3.[15]

Heterogeneity present in doses, injection sites and participant populations led to the decision that a random-effects model should be used for the analysis. RevMan implements a version of random-effects meta-analysis that is described by DerSimonian and Laird[16] and presents an estimate of the between-study variance ($Tau^2$) at the bottom of each forest plot. We tested for statistical homogeneity of pooled estimates of effectiveness using the $X^2$ test and the $I^2$ statistic, for which a statistically significant (p value ≤0.1) value of the $X^2$ test together with $I^2$ value of at least 50% indicates heterogeneity.

Within our eligible comparisons, we split data into migraine classification subgroups in order to show results for chronic migraine, episodic migraine and a mixed group for which the diagnosis could not be split.

We planned to use the following subgroups to test for variation in the effects of the intervention:
1. Trials including medication overuse headache versus trials excluding this type of patient.
2. Different sero-types of BTX (eg, A vs B) and within sero-types (Dysport vs Botox).
3. Different types of agents for the prevention of migraine versus BTX.
4. Accepted and licensed 31 injection pattern versus other injection patterns used.

At least two trials and 200 participants per group were required for any particular subgroup analysis to be carried out.

We carried out sensitivity analyses for our primary outcome only. Prevailing evidence suggests that smaller trials are more likely to report stronger effect estimates than large trials.[17 18] To assess whether these stronger effect estimates reflected the true treatment effect we carried out a sensitivity analysis in which we examined the effect of removing studies at high risk of bias from study size.

We assessed the validity of our findings as well as the level of confidence suitable to any estimates of effect generated by our analyses using the GRADE approach.[19]

## Patient and public involvement

There was no patient or public involvement in the design or reviewing process. However, the final Cochrane manuscript including a lay summary, which is accessible to the public through the Cochrane library, was reviewed by a patient representative as part of the editorial process. Their feedback was incorporated into the final draft.

## RESULTS
### Description of included studies

The flow of information through the review process is given in the Preferred Reporting Items for Systematic Reviews and Meta-Analyses flow chart in online supplementary file 2. The characteristics of studies included in this review are given in online supplementary file 3.

We identified 28 eligible trials, involving a total of 4190 participants, which were eligible for inclusion in this review. Twenty-three of these trials compared BTX-A with placebo injections[6 7 20–40] and three compared with an alternative established oral prophylactic agent.[41–43]

Five trials, reported in four articles, compared alternative doses of BTX-A,[26 33–35] all but one of these also included a placebo arm[26] and one compared with injections of histamine.[44] Due to the paucity of the data, review of the dosing studies and the histamine study are included as appendices in the Cochrane review and is not repeated here.[13]

The results of the critical appraisal were mixed (figure 1). Across all domains poor reporting was an issue and in all but attrition bias and study size at least 50% of trials provided insufficient information to allow

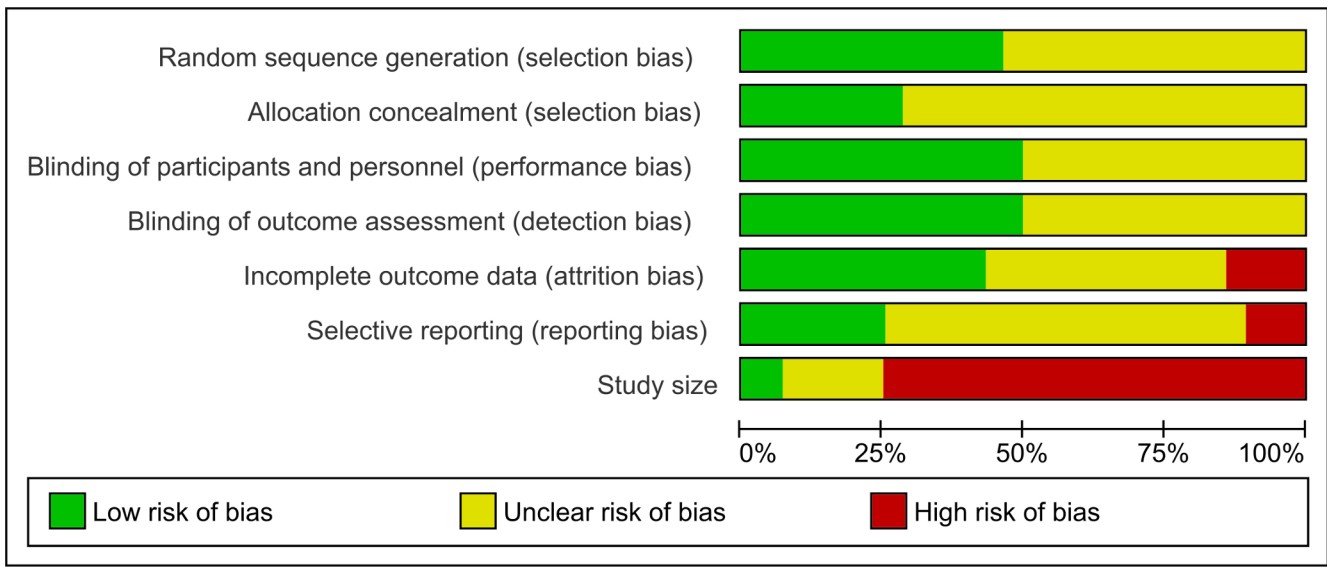

**Figure 1** Risk of bias graph: review authors' judgements about each risk of bias item presented as percentages across all included studies.

judgments about risk of bias to be made. Only two trials were at low risk of bias due to study size (at least 200 participants per trial arm) and these two trials were also at low risk of bias across all other domains.[6 7]

Sixteen trials were commercially sponsored, including the only two trials at low risk from study size.[6 7 21 22 24–27 32–35 40 41 43]

For those trials providing information on the migraine diagnosis of their participants the ratio of chronic/episodic migraine was 1872/1928, leaving 392 included participants unclassified and analysed as' Mixed'. The mean age was 42 years and 85% of all participants were female. Pregnant women were generally explicitly excluded. All included trials used BTX-A, of these 21 had at least one arm treated with the Botox formulation,[6 7 20–24 26 27 30 31 33–35 38 40–44] two used Dysport,[25 32] two used Prosigne[28 31] and one HengLi.[29] The range of doses administered in trials of Botox was 6 U to 300 U. The trials using Dysport administered doses of 80 U up to 240 U in treated arms (dose equivalency reported by trial publications: 2 to 3 U:1 U Botox). HengLi and Prosigne trials used doses ranging from 25 U to 96 U (dose equivalency reported by trial publications: 1 U:1 U Botox).

### Effectiveness versus placebo

Comparison with a placebo group was made in 23 trials with 3912 participants.

Meta-analysis of our primary outcome for the four trials in chronic migraine which reported it showed that there was a reduction of 3.1 days of migraine per month (95% CI −4.7 to −1.4) in favour of BTX-A treatment (figure 2). At least 60% of the participants in this analysis had medication overuse headache. The episodic migraine subgroup involved only one trial of 418 participants which showed no difference in the number of migraine days between treated and placebo groups (p=0.49). Insufficient data were available to carry out any of the planned subgroup analyses on the primary outcome measure. Concern about small trial effects caused us to carry out a sensitivity analysis. Removal of all chronic migraine trials at high risk of bias from study size left just the two PREEMPT trials,

which gave a more conservative reduction of 2.0 days per month (95% CI −2.8 to −1.1, n=1384).

Migraine severity score on a 10 cm visual analogue scale, improved by −3.3 cm (95% CI −4.2 to −2.4) more with active treatment (figure 3). Only four small trials reported meta-analysable data for this outcome. For chronic migraine the improvement was −2.7 cm (95% CI −3.3 to −2.1, n=75), and for episodic migraine it was −4.9 cm (95% CI −6.6 to −3.2, n=34).

A reduction in the number headache days per month of 1.9 days (95% CI −2.7 to −1.0, two trials, n=1384) in favour of BTX-A treatment was also seen. However data for number of migraine attacks from six trials of both chronic migraine and episodic migraine participants (n=2004) showed no significant between group difference (p=0.30). Duration of migraine in hours was fully reported by only one trial showing a greater reduction of −5.1 hours (95% CI −6.2 to −4.0) for 102 chronic and episodic migraine participants. A further four trials with 420 participants reported no significant difference between groups for this outcome. Global assessment measures and quality of life measures were poorly reported and it was not possible to carry out statistical analysis of these outcome measures.

### Effectiveness of BTX versus oral prophylactic agents

Three trials with 178 participants compared Botox injections with oral prophylactic agents using double dummy techniques. Two trials compared 100 U fixed dose plus optional dose of up to 100 U of Botox with topiramate maximum dose 200 mg/day.[42 43] The third trial compared treatment with up to 100 U Botox with sodium valproate 250 mg twice daily.[41] Fourteen of the 178 participants had episodic migraine, all other participants had chronic migraine. Where meta-analysis was possible we pooled data from these three trials as there were insufficient data to allow us to explore comparisons with individual drug types or effects on chronic migraine and episodic migraine populations.

The primary outcome, number of migraine days per month was recorded in only one of the active comparison

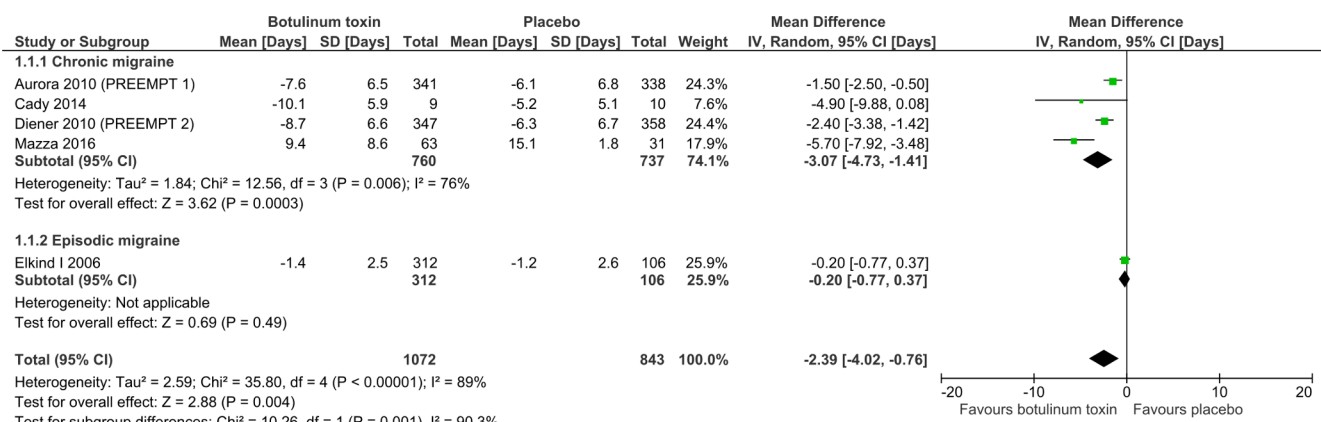

**Figure 2** Comparison of BTX-A versus placebo in relation to number of migraine days per month. BTX-A, botulinum toxin type A.

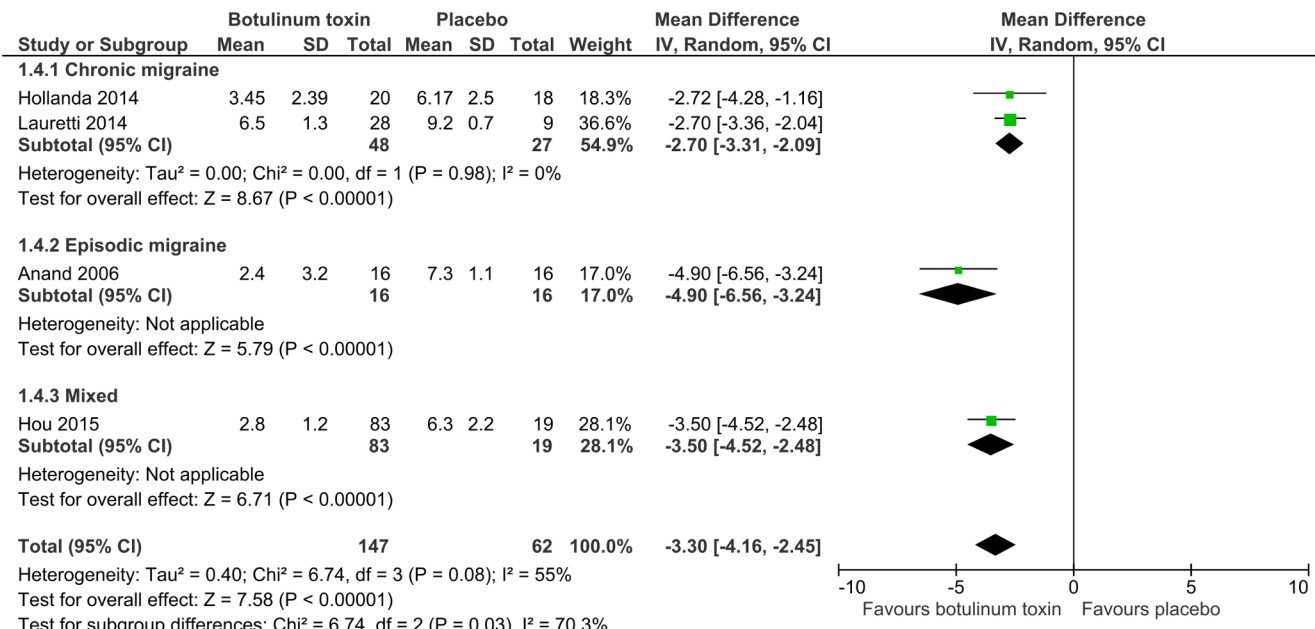

**Figure 3** Comparison of BTX-A versus placebo in relation to severity of migraine measured on a 10 cm visual analogue scale. BTX-A, botulinum toxin type A.

trials. The trialists reported that there was no statistically significant difference between treatment with BTX-A and topiramate for this outcome.[43]

The number of headache days per month was recorded in two trials. No difference in number of headache days per month between treatment with BTX-A and sodium valproate was reported (p=0.55).[35] No data were reported but it was stated that there was also no statistically significant difference between BTX-A and topiramate treated groups.[42] A 5-point scale was used to compare the effect of BTX-A with alternative agents in two trials, Blumenfeld *et al.* reported no significant difference and Mathew *et al.* reported within group analysis only.[41 43] Number

of migraine attacks and duration of migraine were not reported by any trial. No difference between BTX-A and topiramate was stated for use of rescue medications.[43]

Of all the secondary outcome measures, data for meta-analysis were available only for the Migraine Disability Assessment (MIDAS) scores. Results of this showed no significant difference in change scores between the established drug treatments and injection with Botox (p=0.80, two trials, n=101).

## Safety

BTX-A had an RR of treatment related adverse events of twice that seen for placebo (2.2, 95% CI 1.7 to 2.8, six

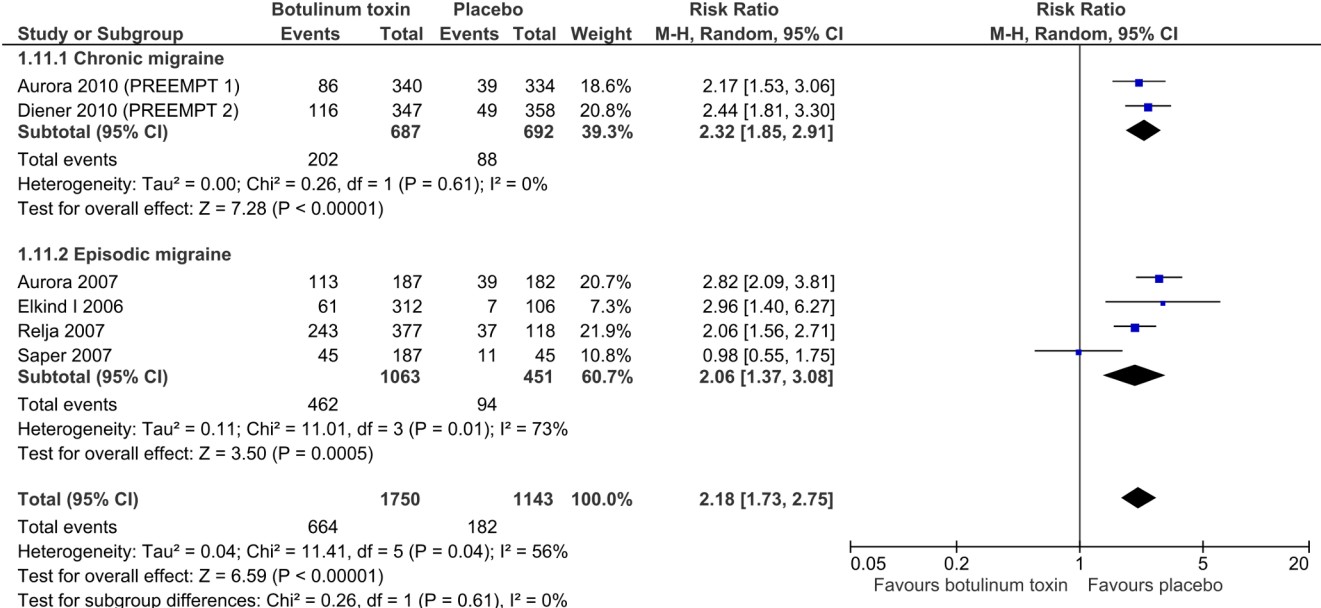

**Figure 4** Comparison of BTX-A versus placebo in relation to treatment related adverse events. BTX-A, botulinum toxin type A.

trials, n=2839) (figure 4). All of these events were transient and non-serious, the most common being blepharoptosis, muscle weakness, injection site pain and neck pain.

Compared with oral treatments, BTX-A showed a reduced RR of treatment related adverse events of 0.76 (95% CI 0.59 to 0.98, two trials, n=73). There was also difference in favour of BTX-A in the RR of withdrawing due to adverse events of 0.28 (95% CI 0.10 to 0.79, $I^2$=0%) which is a RR reduction of 72%.

A low withdrawal rate of 3% for BTX-A was generated using data from all those trials treating with more than one injection cycle irrespective of the type of comparison arm.

### Quality of the evidence

The quality of the evidence assessed using GRADE methods was varied but mostly low and very low; the primary outcome measure was low and very low quality evidence for the placebo and active control comparisons respectively. Small trial size, high risk of bias and unexplained heterogeneity were common reasons for downgrading the quality of the evidence. All judgements and reasons for gradings are given in online supplementary file 4 and 5.

### DISCUSSION

Evidence was identified to support the use of injections of BTX-A into the head and neck muscles, to reduce the number of migraine days experienced per month. Mean frequency of migraine days was significantly reduced by 3 days per month more by BTX-A treatment than by placebo, but this result was revised to 2 days per month as a result of sensitivity analyses. All patients included in this analysis had chronic migraine and so had a high baseline frequency with an average of 20 days per month quoted by the two largest trials in the analysis.[6 7] For patients with chronic migraine, likely to be refractory to first and second line treatment, a 2 to 3 day improvement may well represent a meaningful difference. BTX-A groups also fared better than placebo in the frequency of headache days by 2 days per month. Severity of migraine measured on a visual analogue scale was improved by 3 cm for chronic migraine and 5 cm for episodic migraine on a 10 cm scale. Though these results were from few small trials and the estimate is considered to be low quality evidence, the differences in severity scores were in excess of the minimal clinically important difference of 1.2 cm determined by Kelly[45] and indicate that the treatment may be reducing the impact of each migraine attack. In contrast to this no significant difference from placebo was observed for frequency of migraine attacks. Patient and clinician reported global assessment scales and quality of life scales were underused and when they were incorporated into trials they were poorly reported, so no aggregation of data of this type comparing investigative treatment with placebo was possible in this review.

It was not possible to carry out any analysis on headache diary outcomes or severity measures for head-to-head comparisons between BTX-A and other established agents due to lack of available data. MIDAS scores for 101 patients from two small trials, one comparing Botox with topiramate and one with sodium valproate were available and these showed no significant between group difference (p=0.8).

Trials included in this review commonly state that BTXs have good safety profiles and the evidence from the 23 trials included in this review which reported adverse events in some form support those assertions. Although an increased risk of experiencing treatment related adverse events was found for the BTX-A treated group compared with placebo, the event types were non-serious and transient.

A relative risk reduction (RRR) of 24% in treatment related adverse events in favour of BTX-A was found when comparing with topiramate and sodium valproate in two trials. These two trials found an RRR in favour of BTX-A of 72% for withdrawal rate due to adverse events. Percentage withdrawals due to adverse events for all of those trials included in this review which used more than one round of BTX-A injections, irrespective of the comparison arm type, was 3%. The data sets for the direct comparisons with other prophylactic agents were small, but the relationship is supported by the indirect comparison of this percentage with published rates of 20% for topiramate and 12% for sodium valproate.[46 47] This result suggests that patients tolerate this treatment better than the oral alternatives.

Reporting was generally poor, with only six of 28 trials reporting data on our primary outcome in a usable format, and an additional five providing data for frequency of migraine attacks. These two outcomes are recommended as primary outcomes by the trial guidelines produced by the IHS and should be fully reported to allow individual trials to be placed in the context of the totality of the evidence.[48] A large proportion of the recorded data were missing from the published reports of our included trials. Failure to fully report data in trial publications led to problems throughout the meta-analysis and greater confidence in the conclusions would have been possible if all trials that recorded our outcomes of interest had fully reported them.

Prophylactic treatments for migraine aim to reduce the frequency, duration and/or the intensity of attacks. Frequency of migraine attacks was commonly used as the primary outcome particularly in studies carried out before the publication of the PREEMPT trials. Use of this measure may mask an important improvement in symptoms seen in the form of shorter and less intense migraine attacks. Use of the more sensitive measures, number of days or hours spent with migraine per month coupled with a measure of intensity, may enable detection of such changes and could be particularly relevant to episodic migraine patients for whom attacks may be shorter at baseline. Another problem with focusing on

this outcome measure was the failure generally to define what was meant by a migraine attack, and therefore, the likelihood of variation in the definitions used across the trials.

Neither efficacy nor safety data were available for long-term treatment with BTX. The longest treatment period in any of the studies included in this review was three treatments with 12 weeks between treatments, so we cannot know the implications of treating patients with BTX over a period longer than 9 months.

Most trials did not report whether or not they had included patients with medication overuse symptoms and those that did stated they had largely excluded medication overuse patients. Pooled data for the two PREEMPT trials for the chronic migraine plus medication overuse subgroup (n=906) showed that the difference between groups for both migraine and headache day frequencies was 2 days (p<0.001) in favour of treatment with BTX.[49] The medication overuse subgroup result falls within the CIs of the pooled estimate generated by this review for the same outcome measure in combined populations with and without medication overuse headache. It would appear from these data that the inclusion of patients with medication overuse does not change the effectiveness of BTX for prophylactic treatment of migraine.

## CONCLUSIONS

We have data which suggest that BTX effectively reduces the duration and severity of migraines in sufferers. There are however question marks over the quality of the evidence. Efficacy measures were commonly reported as showing non-inferiority of BTX to topiramate and sodium valproate and the withdrawal rate from BTX is much lower than that for first line prophylactic treatments for migraine. So should we be using more BTX?

It is currently recommended by NICE guidance that medication overuse headache should be addressed before treatment with BTX but trial data suggests it is efficacious in chronic migraine patients with untreated medication overuse headache. So although treatment of medication overuse headache is good practice, perhaps it should not be a requirement before prescription of BTX. NICE recommends the use of BTX to treat chronic migraine that has not responded to at least three prior pharmacological prophylaxis therapies. The confidence in the effectiveness of these drugs is arguably no greater than that for BTX and patients seem better able to tolerate BTX.[4 5 46 47] If, as is suggested by trial data, BTX has the equivalent efficacy to other agents but lower withdrawal rates, then if it were not for the higher cost, BTX would likely be recommended as an earlier preventative treatment for chronic migraine.

The difference between chronic and episodic migraine diagnoses is arbitrary and so there is no pathophysiological reason that treatment with BTX would be efficacious in people with 15 days of headache per month and inefficacious in people with 14 days of headache per month in a stepwise fashion. The treatment may well be useful for episodic migraine, particularly in high frequency episodic migraine, but data is lacking.

**Contributors** C Clarke and A Sinclair conceived the review. C Herd ran searches not covered by the review group's information specialist. C Herd, C Tomlinson, C Rick and A Sinclair screened the search results. C Herd, C Tomlinson, C Rick, W Scotton, C Clarke and A Sinclair assessed the quality of studies and extracted data. C Herd contacted trial authors, managed data and entered it into RevMan, and carried out data analysis. N Ives provided statistical advice. C Herd, C Tomlinson, C Rick, N Ives, C Clarke and A Sinclair were involved in interpretation of the results. All authors read and edited final version of the review.

**Funding** This research received no specific grant from any funding agency in the public, commercial or not-for-profit sectors.

**Competing interests** Yes, there are competing interests for one or more authors and I have provided a Competing Interests statement in my manuscript.

**Provenance and peer review** Not commissioned; externally peer reviewed.

**Data sharing statement** Full data analysis and trial summaries available in Cochrane review DOI: 10.1002/14651858.CD011616.pub2.

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
