## [Reviewer comments · BMJ Open]

ARTICLE DETAILS

TITLE (PROVISIONAL)	Cochrane systematic review and meta-analysis of Botulinum toxin for the prevention of migraine.
AUTHORS	Herd, Clare (proxy) (contact); Tomlinson, Claire; Rick, Caroline; Scotton, William; Edwards, Julie; Ives, Natalie; Clarke, Carl; Sinclair, AJ

VERSION 1 – REVIEW

REVIEWER	Stephen Silberstein, MD Thomas Jefferson University USA
REVIEW RETURNED	30-Jan-2019

GENERAL COMMENTS	Thank you for doing this.
---------------------------

REVIEWER	John Ohrvik Uppsala University, Sweden
REVIEW RETURNED	17-Feb-2019

GENERAL COMMENTS	This is well performed meta-analysis evaluating the effects of botulinum toxin for prevention of migraine in adults. The authors also discuss the relative risk of adverse events when comparing botulinum toxin with placebo and oral treatments. From a statistical point of view, I have the following comments: 1. I suggest that the authors instead of writing that this meta-analysis shows that botulinum toxin resulted in reduced frequency of -3.1 migraine days/month in chronic migraineurs compared to placebo present the more conservative reduction of 2.0 days/month, by removing the two small trials with very high reduction (-4.9 and -5.7 days/month) which probably are due to publication bias.2. The random effect model used in case of significant heterogeneity has to be better explained since the weighting could be done in different ways. Further-more Tau has to be defined.3. The chi-square approximation of Cochran's Q statistics under the null hypothesis – no heterogeneity among the studies – is not accurate for small and moderate study sizes. Kulinskaya et al (Biometrics 67, 203-212, March 2011) suggest a method based on fractional degrees of freedom (df) which substantially improve the original chi-square distribution with $s-1$ df, where s is the number of studies included in the meta analysis.4. Migraine severity was assessed by a 10 cm VAS. These
--

	data were analyzed using the normal distribution approximation. I suggest that the authors either use rank based methods eg the Wilcoxon Mann-Whitney rank sum test and corresponding 95% confidence interval or check the normality of the VAS data.
--	---

REVIEWER	Tamara Pringsheim University of Calgary, Canada
REVIEW RETURNED	25-Feb-2019

GENERAL COMMENTS	This is a summary of a published Cochrane review on the use of botulinum toxin injections for migraine headache. Methodologically the systematic review has been performed rigorously according to the Cochrane standard. The presentation of results is clear, as is the discussion and conclusions of the paper. On the whole, this paper is relevant to clinicians, given how common migraine is and the therapeutic options available, and is scientifically well done. The only thing to mention is that I did not see any mention of assessment for publication bias, which is traditionally performed in Cochrane reviews if enough studies are found.
---

REVIEWER	Yohannes W Woldeamanuel Stanford University, USA Advanced Clinical and Research Center, Ethiopia
REVIEW RETURNED	27-Feb-2019

GENERAL COMMENTS	This manuscript contains highly heterogeneous data from different studies with variable methodologies. This renders it in combinable for conducting a meta-analysis, as can be seen by the high levels of statistical heterogeneity. Most of the meta-analysis are based on mere 2 studies; achieving low sample size and power for meta-analysis. I recommend the authors train on how to power adequate sample size and study power for running meta-analysis. I am including below a couple of citations for that. Sample size and study power for meta-analysis is not done a priori. How is statistical power for meta-analysis computed? Particularly when using random-effects weighted analysis. - Jackson D, Turner R. Power analysis for random-effects meta-analysis. Res Syn Meth. 2017;8:290–302. https://doi.org/10.1002/jrsm.1240 Valentine, J. C., Pigott, T. D. & Rothstein, H. R. (2010). How many studies do you need? A primer on statistical power for meta-analysis. Journal of Educational and Behavioral Statistics, 35(2), 215-247.\n– Chapters 4 -6 in Pigott, T. D. (2012). Advances in meta-analysis. New York, NY: Springer - It is impossible to even stratify the 2-3 available studies to different confounders such as EM vs CM or MOH presence or absence, inferiority trials vs placebo comparisons... - This manuscript contains major flaws that cannot be rectified. Hence, I recommend to reject it.
---

VERSION 1 – AUTHOR RESPONSE

Reviewer(s)' Comments to Author:

Reviewer: 1

Reviewer Name: Stephen Silberstein, MD

Institution and Country: Thomas Jefferson University, USA

Please state any competing interests or state 'None declared': None

Please leave your comments for the authors below Thank you for doing this

Reviewer: 2

Reviewer Name: John Ohrvik

Institution and Country: Uppsala University, Sweden

Please state any competing interests or state 'None declared': None declared

Please leave your comments for the authors below This is well performed meta-analysis evaluating the effects of botulinum toxin for prevention of migraine in adults. The authors also discuss the relative risk of adverse events when comparing botulinum toxin with placebo and oral treatments. From a statistical point of view, I have the following comments:

I suggest that the authors instead of writing that this meta-analysis shows that botulinum toxin resulted in reduced frequency of -3.1 migraine days/month in chronic migraineurs compared to placebo present the more conservative reduction of 2.0 days/month, by removing the two small trials with very high reduction (-4.9 and -5.7 days/month) which probably are due to publication bias.

Abstract results changed to 2 days.

Results unchanged as both estimates were discussed.

Discussion changed to : Mean frequency of migraine days was significantly reduced by 3 days per month more by BTX type-A treatment than by placebo, but this result was revised to 2 days per month as a result of sensitivity analyses. All patients included in this analysis had chronic migraine and so had a high baseline frequency with an average of 20 days per month quoted by the two largest trials in the analysis. For patients with chronic migraine, likely to be refractory to first and second line treatment, a 2-3 day improvement may well represent a meaningful difference.

The random effect model used in case of significant heterogeneity has to be better explained since the weighting could be done in different ways. Further-more Tau has to be defined.

Added the following text : RevMan implements a version of random-effects meta-analysis that is described by Dersimonian and Laird(reference) and presents an estimate of the between-study variance (Tau²) at the bottom of each forest plot.

The chi-square approximation of Cochran's Q statistics under the null hypothesis - no heterogeneity among the studies - is not accurate for small and moderate study sizes. Kulinskaya et al (Biometrics 67, 203-212, March 2011) suggest a method based on fractional degrees of freedom (df) which

substantially improve the original chi-square distribution with $s-1$ df, where s is the number of studies included in the meta analysis.

We have followed Cochrane methodology as the analysis was originally part of a Cochrane review. Cochrane recognises the limitations of the chi square assumption for Q in small and moderately sized trials and address this by suggesting use of a P value of 0.10 rather than the conventional level of 0.5 to account for this and even beyond that they suggest heterogeneity should be confirmed by a P value lower than 0.1 but not ruled out because the value is higher than 0.1.

Text in review: for which a statistically significant (P value ≤ 0.1) value of the Chi2 test together with I2 value of at least 50% indicates heterogeneity.

Thank you for the reference will we will refer to for future projects.

Migraine severity was assessed by a 10 cm VAS. These data were analyzed using the normal distribution approximation. I suggest that the authors either use rank based methods eg the Wilcoxon Mann-Whitney rank sum test and corresponding 95% confidence interval or check the normality of the VAS data.

As we only have access to aggregate data reported in trial publications, in all cases as means and SDs, we cannot re-analyse in this way. If the data was skewed we would expect it to have been reported by trialists in a different format.

Reviewer: 3

Reviewer Name: Tamara Pringsheim

Institution and Country: University of Calgary, Canada

Please state any competing interests or state 'None declared': None declared

Please leave your comments for the authors below

This is a summary of a published Cochrane review on the use of botulinum toxin injections for migraine headache. Methodologically the systematic review has been performed rigorously according to the Cochrane standard. The presentation of results is clear, as is the discussion and conclusions of the paper. On the whole, this paper is relevant to clinicians, given how common migraine is and the therapeutic options available, and is scientifically well done.

The only thing to mention is that I did not see any mention of assessment for publication bias, which is traditionally performed in Cochrane reviews if enough studies are found.

Text from COCHRANE REVIEW: We considered the use of funnel plots to assess the risk of publication bias but did not carry them out. We made this decision because of the small number of trials included in the individual meta-analyses and the true heterogeneity in the trial design (dose, injection paradigm) and populations studied (migraine subclassifications), which would make it impossible to draw useful conclusions from the plots.

Instead we relied upon the GRADE process which incorporates assessed of the risk of publication bias into its ratings. The GRADE tables are given as supplemental files for this submission.

Reviewer: 4

Reviewer Name: Yohannes W Woldeamanuel

Institution and Country: Stanford University, USA, Advanced Clinical and Research Center, Ethiopia

Please state any competing interests or state 'None declared': None declared.

Please leave your comments for the authors below This manuscript contains highly heterogeneous data from different studies with variable methodologies. This renders it in combinable for conducting a meta-analysis, as can be seen by the high levels of statistical heterogeneity. Most of the meta-analysis are based on mere 2 studies; achieving low sample size and power for meta-analysis. I recommend the authors train on how to power adequate sample size and study power for running meta-analysis. I am including below a couple of citations for that. Sample size and study power for meta-analysis is not done a priori. How is statistical power for meta-analysis computed? Particularly when using random-effects weighted analysis.

- Jackson D, Turner R. Power analysis for random-effects meta-analysis. *Res Syn Meth.* 2017;8:290-302. <https://doi.org/10.1002/jrsm.1240> Valentine, J. C., Pigott, T. D. & Rothstein, H. R. (2010). How many studies do you need? A primer on statistical power for meta-analysis. *Journal of Educational and Behavioral Statistics*, 35(2), 215-247.\n- Chapters 4 -6 in Pigott, T. D. (2012). *Advances in meta-analysis*. New York, NY: Springer

- It is impossible to even stratify the 2-3 available studies to different confounders such as EM vs CM or MOH presence or absence, inferiority trials vs placebo comparisons...

- This manuscript contains major flaws that cannot be rectified. Hence, I recommend to reject it.

We have followed approved Cochrane methods and were under scrutiny throughout the reviewing process by their editors and reviewers. We comment on the effect of small studies and investigate those with a sensitivity analysis. The paper referenced above does not indicate that the random effects model should not be used but merely that the results of it should be interpreted with caution, which we feel we have done. It also acknowledges the practical difficulties in carrying out an a priori power calculation for a systematic review and this is not common practice. In addition our protocol predates the publication of this article by two years.

As we are dealing with RCT's the characteristics you mention are not considered to be confounders, but we listed them as potential subgroups in our protocol (at which point we did not know how many trials we would retrieve), for use if we had sufficient data. Ultimately the feedback from headache specialists was that the forest plots were more meaningful if data for EM and CM were separated, we did not use any other subgroupings as the dataset did not allow it.

VERSION 2 – REVIEW

REVIEWER	John Ohrvik Uppsala University, Sweden
REVIEW RETURNED	21-Apr-2019
GENERAL COMMENTS	The revised manuscript has improved. The comments and queries have been addressed in a satisfactory manner. I think this paper could be of value for clinicians.

VERSION 2 – AUTHOR RESPONSE

Reviewer: 2

Reviewer Name: John Ohrvik

Institution and Country: Uppsala University, Sweden

Please state any competing interests or state 'None declared': None declared

Please leave your comments for the authors below

The revised manuscript has improved. The comments and queries have been addressed in a satisfactory manner. I think this paper could be of value for clinicians.